# QTLs and Candidate Loci Associated with Drought Tolerance Traits of Kaybonnet x ZHE733 Recombinant Inbred Lines Rice Population

**DOI:** 10.3390/ijms242015167

**Published:** 2023-10-14

**Authors:** Yheni Dwiningsih, Julie Thomas, Anuj Kumar, Chirag Gupta, Navdeep Gill, Charles Ruiz, Jawaher Alkahtani, Niranjan Baisakh, Andy Pereira

**Affiliations:** 1Department of Crop, Soil, and Environmental Sciences, Faculty of Agriculture Food and Life Sciences, University of Arkansas System Division of Agriculture, Fayetteville, AR 72701, USA; ydwining@uark.edu (Y.D.); jt008@uark.edu (J.T.); axk018@uark.edu (A.K.); ceruiz@uark.edu (C.R.); jsalqahtani@ksu.edu.sa (J.A.); 2Waisman Center, University of Wisconsin-Madison, Madison, WI 53705, USA; cgupta8@wisc.edu; 3Department of Biostatistics and Medical Informatics, University of Wisconsin-Madison, Madison, WI 53706, USA; 4Department of Biological Sciences, Nova Southeastern University, Fort Lauderdale, FL 33314, USA; ngill@nova.edu; 5Department of School of Plant, Environmental and Soil Sciences, Louisiana State University, Baton Rouge, LA 70803, USA; nbaisakh@agcenter.lsu.edu

**Keywords:** rice, drought stress, QTLs, candidate genes, gene expression

## Abstract

Rice is the most important staple crop for the sustenance of the world’s population, and drought is a major factor limiting rice production. Quantitative trait locus (QTL) analysis of drought-resistance-related traits was conducted on a recombinant inbred line (RIL) population derived from the self-fed progeny of a cross between the drought-resistant *tropical japonica* U.S. adapted cultivar Kaybonnet and the drought-sensitive *indica* cultivar ZHE733. K/Z RIL population of 198 lines was screened in the field at Fayetteville (AR) for three consecutive years under controlled drought stress (DS) and well-watered (WW) treatment during the reproductive stage. The effects of DS were quantified by measuring morphological traits, grain yield components, and root architectural traits. A QTL analysis using a set of 4133 single nucleotide polymorphism (SNP) markers and the QTL IciMapping identified 41 QTLs and 184 candidate genes for drought-related traits within the DR-QTL regions. RT-qPCR in parental lines was used to confirm the putative candidate genes. The comparison between the drought-resistant parent (Kaybonnet) and the drought-sensitive parent (ZHE733) under DS conditions revealed that the gene expression of 15 candidate DR genes with known annotations and two candidate DR genes with unknown annotations within the DR-QTL regions was up-regulated in the drought-resistant parent (Kaybonnet). The outcomes of this research provide essential information that can be utilized in developing drought-resistant rice cultivars that have higher productivity when DS conditions are prevalent.

## 1. Introduction

A model species for monocot and cereal plants, rice (*Oryza sativa* L.) has a compact diploid genome size of about 500 Mb and 12 chromosomes, making it one of the rich sources of carbohydrates and commercially productive cereal crops that serves as the primary food source for about 2.5 billion people worldwide [1]. The rice crop study with a small, well-annotated genome provides an excellent tool for this research to identify chromosome regions for drought resistance. The primary rice-producing countries are China, India, Indonesia, Bangladesh, Vietnam, Thailand, Myanmar, Philippines, Brazil, Japan, the United States, Pakistan, and the Republic of Korea [2]. Additionally, Arkansas is the nation’s top producer of long- and medium-grain rice, and the United States is the third-largest exporter of rice overall [3].

According to numerous statistical projections and polls, a 30% increase in world-wide rice production is required over the next 20 years to meet the economic growth and growing food demands of the global population [4]. Currently, the rate of increase in food production is outpacing the rate of population growth in the world. In addition, drought has become the most crucial constraint in rice production due to global climate change and the competition with urban and industrial users for the limited water available [5,6]. Approximately 50% of the total global rice area is affected by drought. Around 130 million ha of rice fields in Asia are annually affected by drought, and droughts are predicted to become more frequent in many regions in the future [7]. Compared to current drought levels, future droughts could reduce rice production even more [8]. Drought stress mainly affects the rice crop plant at the physiological, morphological, and molecular level [9]. The majority of rice growth stages—seedling, vegetative, and reproductive—are affected by drought, with the reproductive stage being the most sensitive to DS conditions [10,11,12]. Moreover, DS conditions at the reproductive stage cause a significant reduction in the grain yield components such as spikelet per panicle number, panicle length, primary panicle branch number, filled grain per panicle number, and hundred grain weight, all leading to a decrease in grain yield per plant [13].

Rice crops require three times as much water as other cereal crops such as maize and wheat in order to produce just one kilogram of the final product [14]. It has been suggested that exposure to DS during the reproductive phase can decrease grain yields by as much as 77% [9]. Annually, DS conditions cause a loss of roughly 18 million tons of rice. However, developing drought-resistant rice genotypes that perform better under DS conditions can be aided by using cutting-edge genomic technologies in combination with high-quality rice genome sequence information. Several drought-resistant rice genotypes, including Vandana, Nagina-22 (N22), Bengal, and Kaybonnet, are being used in a drought-resistance molecular breeding program to study the identification of quantitative trait loci (DR-QTLs) regulating grain yields under DS conditions [15,16,17]. Increased yield output and stable food supply can be achieved through the development of a drought-resistant rice genotype.

Quantitative trait loci (QTLs) are regions of DNA that encode genes for quantitative traits, the effects of which can be measured quantitatively. Plant height, grain yield, and susceptibility to abiotic and biotic stress are all examples of quantitative traits that are controlled by one or more genes and affected by environmental variation. Molecular markers are used to define and map QTLs. Recent advances in genomic technology and statistical analysis methodologies have made this mapping approach particularly cost-effective for plant research projects [18,19]. SNPs, or single-nucleotide polymorphisms, are a common type of DNA marker used to locate QTLs for desirable traits in plants. These SNPs are the most prevalent type of variation in rice genomes and are crucial for high-resolution genotyping and generating the most accurate maps possible [20].

In addition, single nucleotide polymorphisms are both productive and economical [1,21]. As a result of the advent of genotyping-by-sequencing (GBS) technology, single nucleotide polymorphisms have emerged as the most widely used DNA markers in the 21st century [22]. In order to collect genomic information regarding a wide variety of agronomic traits, QTL analysis has been utilized. As a result, genomic information obtained from QTL analysis might be useful for improving plant breeding programs using a technique known as marker-assisted selection (MAS).

In the past few decades, QTL mapping for agronomic and physiological traits under abiotic stress conditions has involved in the improvement of rice production. A number of QTLs associated with drought-resistant (DR) traits have been identified in *Oryza sativa* L. [23]. Furthermore, DR-QTL mapping has been very useful in identifying the genes and chromosomal segments associated with complex DR traits. Drought-resistant rice genotype development has been slow due to the genetic complexity that influences grain yield features under drought stress and the lack of rice crop growth models to analyze the yield gap under limited water resources [24]. However, several studies at IRRI have reported that the development of mapping populations derived from a drought-resistant variety and a high-yielding variety has proven effective in combining drought resistance with high-yield potential. These mapping populations have also shown transgressive segregants with higher yields compared to the parents under drought stress and normal conditions.

A number of QTLs associated with drought-resistant (DR) traits have been identified in *Oryza sativa* [23]. In 1996, QTLs for osmotic adjustment were mapped by Lilley et al. [25]. The QTL for leaf rolling in the double haploid population of IR64/Azucena under drought stress were identified by Courtois et al. [26]. Price et al. [27,28], mapped QTL for leaf rolling under drought stress on chromosome 1 by using an F2 population. QTLs for tiller number, plant height, total root number, root–shoot ratio, and root dry weight were identified by Kanbar et al. [29] in the double haploid population of CT9993/IR62266 under well-watered conditions. QTLs linked to leaf length, tiller number, and nitrogen content were identified by Xu et al. [30] in the backcross inbred line (BIL) population derived from a cross between *temperate japonica* and *aus*. In 2010, Gomez et al. [31] also detected QTL for leaf rolling and leaf drying on chromosome 1 in the RIL population. Furthermore, DR-QTL mapping has been very useful in identifying the genes and chromosomal segments associated with complex DR traits.

Several techniques, such as expressed sequence tag (EST) profiling, transcript profiling via massively parallel signature sequencing (MPSS), microarrays and quantitative real-time PCR, RNA gel blot analysis [32], and comparative proteome analysis [33], have been used to identify multiple genes associated with drought stress resistance in rice. However, few genes have been functionally validated for their drought resistance in rice [34]. Important examples include the stress-responsive rice NAC genes SNAC1, OsNAC6/SNAC2, and OsNAC5, which, when over-expressed, increase drought resistance [35].

By controlling stomatal closure and thereby reducing water loss through transpiration, the plant hormone abscisic acid (ABA) plays a crucial role in adaptive responses to drought stress [36,37]. Drought stress causes a rise in endogenous ABA concentrations, which aids plant adaptation to the water deficit [38]. Many genes involved in a plant’s defense mechanism are regulated by ABA, making them more or less sensitive to drought [39]. ABA-responsive gene expression is controlled by a number of transcription factors, including ABFs/AREBs [40], CBF/DREB, MYB, NAC, and WRKY [41].

The identification of genomic regions related to the expression of complex drought traits using recombinant inbred line (RIL) populations, which contain many gene recombinations, and subsequent validations of the associated loci with gene expression using additional studies of genomes from a diverse panel representing six distinct subpopulations for the role of SNPs regulating the expression of the QTL genes are among the few recent studies on drought tolerance. This study provides extensive phenotypic data associated with the RIL population to find drought-resistant QTLs and candidate genes in the K/Z RIL population under both drought stress and well-watered conditions by examining changes in the morphological traits, grain yield components, imposed drought conditions, and root architectural traits linked to the ABA response. Gene expression analysis was used to further validate the QTLs that showed high logarithm of odd (LOD) and phenotypic variation explanation (PVE) in both the tolerant and sensitive parental lines. The identification of QTLs for drought-related traits in the two contrasting cultivars will contribute to our understanding of the genetic control of rice productivity at the sensitive reproductive stage under drought-stress conditions, which in turn will accelerate the development of drought-resistant rice varieties with improved grain yields under drought stress conditions.

## 2. Results

### 2.1. Analysis of Parents K/Z RIL Population–Drought Resistant and Drought Sensitive

The variations in the root anatomy and root morphology of drought-resistant Kaybonnet and drought-sensitive ZHE733 are presented in Figure 1 and Appendix A, respectively. Kaybonnet displayed a root that was longer than ZHE733. In the vascular stele cross-section, the midsection of primary root of the drought-resistant parent Kaybonnet enclose two metaxylem numbers, compared to only one metaxylem for the drought-sensitive parent ZHE733. Additionally, the overall aerenchyma area of Kaybonnet measures 0.34 mm^2^, which is larger than that of ZHE733 (0.29 mm^2^). According to Zhu et al.’s [42] findings, aerenchyma improves a plant’s tolerance to drought by lowering the metabolic cost of the roots and increasing the plant’s ability to draw water from dry soil. In conclusion, the root anatomical traits contributed to greater drought resistance in parent Kaybonnet when compared to parent ZHE733.

Based on the screening of grain yield in DS conditions at the reproductive stage (R3), Kaybonnet is the more drought-resistant of the two parental genotypes, while ZHE733 exhibits a phenotype indicative of sensitivity to drought. Under DS conditions, Kaybonnet showed a 20% reduction in filled grains per panicle, while ZHE733 showed a 50% decrease in grain filling (Figure 2). The progeny of a cross between genotypes that are resistant or sensitive to drought can be used to examine the inheritance of drought resistance and to identify key QTLs that affect grain yields under DS conditions [43,44]. A total of 198 K/Z RILs were examined for the number of filled grains per panicle; of these, 13.1 percent were found to be highly drought-resistant lines, 11.1 percent to be moderately drought-resistant lines, and 75.7 percent to be sensitive to drought (Appendix A). This suggests that a number of factors may be at play in the inheritance of drought-resistant and drought-sensitive phenotypes in the population.

### 2.2. Variation in Morphological Traits of K/Z RILs under Drought Stress Conditions

A normal frequency distribution (Figure 3 and Figure 4) in the RIL population indicated quantitative inheritance for morphological features such plant height (PH), tiller number (TN), heading days (HD), flag leaf width (FLW), and drought leaf rolling (DLR); therefore, these traits in the progeny are suitable for QTL analysis [45]. For plant height and tiller number, there were many morphological features within the RIL population that varied between WW and DS conditions, while HD, FLW, and DLR measurements under DS conditions displayed a transgressive segregation pattern.

### 2.3. Genetic Variation for Grain Yield Components under Reproductive-Stage Drought Stress

The total plant biomass (TPB), spikelet per panicle number (SP), filled grain per panicle number (FG), panicle length (PL), and primary panicle branch number (PPB) mean values of the grain yield components revealed a significant difference in drought-tolerance for both parents (KB and ZHE 733) and K/Z RIL population under DS conditions compared to WW (Table 1). Under WW condition, KB-WW (14.80 g) and ZHE733-WW (13.4 g) did not show a significant difference in total plant biomass, while under drought stress, the KB-DS (14.10 g) showed a significant decrease compared to ZHE733-DS (6.50 g), suggesting severe reductions in ZHE733 (51.49%) than KB (4,73%) under DS. The spikelet per panicle number (SP) did not show significant differences under WW conditions. For KB-WW (104) and ZHE733-WW (90), while under drought stress, the spikelet per panicle number KB-DS (74) showed a significant decrease compared to ZHE733-DS (55), suggesting more of a reduction in ZHE733 (38.89%) than KB (28.85%) under DS.

Under WW conditions, KB-WW (66) and ZHE733-WW (43.2) did not show a significant difference in the filled grain per panicle number; while under drought stress, the KB-DS (50) showed a significant decrease compared to ZHE733-DS (16), suggesting a more severe reduction in ZHE733 (62.96%) than KB (24.24%) under DS. Under WW conditions, KB-WW (21.7 cm) and ZHE733-WW (18.6 cm) did not show a significant difference in panicle length; while under drought stress, the KB-DS (19.3 cm) showed a significant decrease compared to ZHE733-DS (15.6 cm), suggesting a more severe reduction in ZHE733 (16.13%) than KB (11.06%) under DS. The primary panicle branch number showed significant differences under WW conditions between KB-WW (19) and ZHE733-WW (8); while under drought stress, the KB-DS (13) showed significant decreases compared to ZHE733-DS (4), suggesting a more severe reduction in ZHE733 (50%) than KB (31.58%) under DS.

The phenotypic trait TPB under WW showed normal distribution; while under DS, the overall RIL population was better performing than parent KB. The SPP showed normal distribution under WW, but under DS, the trait showed transgressive segregation. The RIL population was more weighted towards ZHE733 parent for trait FG under both WW and DS. The trait PL has a normal for RIL population under both WW and DS. Based on PPB, the RIL population was distributed between both parents under both WW and DS. When comparing the average values of traits in RIL populations, there is a reduction under DS compared to WW for following traits: TPB (12.5%), SP (38.75%), FG (47.62%), PL (15.82%), and PPB (16.92%).

### 2.4. Variation in Root Architectural Traits under ABA Conditions

The responses of the parents, Kaybonnet and ZHE733, to ABA varied; Kaybonnet, a parent that is drought-resistant, displayed greater sensitivity to ABA than ZHE733, a parent that is drought-sensitive. Under ABA conditions compared to control conditions, both parents showed reduced root length (RL), total root number (TRN), shallow root number (SRN), deep root number (DRN), and root fresh weight (RFW). In contrast to ZHE733, Kaybonnet showed a greater reduction in RL, TRN, SRN, DRN, and RFW (Appendix A). Under ABA conditions (3 µM), Kaybonnet showed a significant reduction in RL, TRN, SRN, DRN, and RFW, with respective values of 61.94%, 62.12%, 100%, and 66.22%. Under ABA conditions (3 µM), ZHE733 displayed a lesser reduction in RL, TRN, SRN, DRN, and RFW, with respective values of 17.59%, 5.43%, 4.68%, 5.79%, and 6.45%. Additionally, in comparison to Kaybonnet, ZHE733 showed less reduction in RL, TRN, SRN, DRN, and RFW under ABA conditions (5 µM).

The K/Z RIL population showed variations in their root architectural traits in response to ABA (0 µM, 3 µM, and 5 µM), such as reduction in RL (Appendix A), increase in RSR (Appendix A), reduction in TRN (Appendix A), decrease in SRN (Appendix A), shortening in DRN (Appendix A), and lessening RFW (Appendix A). Additionally, a nearly normal distribution was seen in the distribution of the root architectural traits examined under ABA conditions. The K/Z RIL population’s range of RL, RSR, TRN, SRN, DRN, and RFW under ABA conditions likewise showed a great deal of diversity as morphological traits (Appendix A).

In addition, the average root architectural traits of the RILs lie between the parents under ABA conditions. The average RL, TRN, SRN, and RFW in the RIL population under ABA conditions 3 µM all showed a reduction, with 18.69%, 16.46%, 33.78%, 9.26%, and 2.94%, respectively. In addition, when the concentration of ABA was increased to 5 µM, the average levels of RL, TRN, SRN, and RFW in the RIL population all decreased by 21.16%, 21.07%, 36.51%, and 12.61%, respectively. Despite the fact that the RL, TRN, SRN, DRN, and RFW of both parents and the RIL population decreased in response to ABA conditions, the reduction in Kaybonnet was significantly greater. The root–shoot ratio (RSR) increased by 38% and 16% for Kaybonnet under ABA conditions 3 µM and 5 µM; for the RIL population under ABA conditions, 3 µM and 5 µM increased 22.92% and 22.69%, respectively.

### 2.5. Correlation of Morphological Traits and Grain Yield Components under WW and DS Conditions with Root Architectural Traits under ABA Conditions

According to Gibert et al. [46], correlation analysis assists in improving the understanding of the broad impact that various rice plant attributes have to each other. Correlation analysis increases an understanding of the overall contribution of various rice plant traits to each other [46]. A Pearson’s correlation coefficient analysis was conducted on the morphological traits and grain yield components under WW and DS environments, in addition to root architectural traits under ABA conditions, with the purpose of determining the degree to which these traits are related to one another (Appendix A). All of the characteristics under investigation were shown to have statistically significant connections with one another. 

Three RIL lines (100007, 100036, and 100135) showed drought resistance based on morphological traits (PH and TN), grain yield components (TPB, SP, FG, PL, and PPB), and ABA sensitivity response characteristics similar to Kaybonnet as the drought-resistant parent. Under DS conditions, these three groups showed a decrease of 30% or less in PH, TN, TPB, SP, FG, PL, and PPB. Meanwhile, these three lines showed greater than a 50% reduction in root architectural traits (RL, TRN, SRN, DRN, and RFW). In addition, these three lines showed short PH and high numbers in TN, TPB, SP, PL, FG, and PPB based on the morphological features and grain yield components under WW conditions, making them a valuable genetic resource for the development of drought-resistant varieties. In WW conditions, values for PH, TN, TPB, SP, PL, FG, and PPB ranged from 42 to 51 cm, 5 to 9, 17 to 24 g, 104 to 130, 17.5 to 23.7 cm, 100 to 103, and 10 to 12, respectively.

### 2.6. Relationships of Root Anatomy and Drought Resistance in Tolerant and Sensitive Lines

In comparison to ZHE733 and drought-sensitive lines, parental Kaybonnet and drought-resistant lines displayed more metaxylem vessels (Figure 5). According to Figure 1, Kaybonnet has two metaxylem vessels, but ZHE733 and drought-sensitive lines only have one. Drought-resistant lines typically have four metaxylem vessels. Drought-resistant lines only have a metaxylem size of 0.0005 mm^2^, while drought-sensitive lines have an average metaxylem size of 0.0008 mm^2^. In drought-resistant lines, aerenchyma typically covers 60% of the total root cross section, compared to only 40% in drought-sensitive lines. In drought-sensitive lines, there are generally 11 cortical cell files, which is more than the average of 5 in drought-resistant lines.

### 2.7. High-Density Genetic Linkage Map with GBS Markers

Understanding the plant genome and learning about allele introgression through plant breeding are both aided by genetic linkage maps [47]. More precise candidate genes for gene cloning and subsequent validation with reverse genetics procedures can be predicted by employing high-density genetic linkage map results [48]. A total of 200 samples (2 parental lines and 198 RILs) were analyzed using GBS, yielding 28,598 SNP markers with a heterozygosity level of 1.3% and non-parental alleles at 0.4%. SNP markers were filtered based on the fraction of missing data (90%), minor allele frequency (MAF 1%), parental polymorphism, recombinant frequency, and heterozygosity. QTL IciMapping software version 4.2.53 with the Kosambi mapping function was used to create the high-density genetic linkage map, and 4133 filtered SNP markers were also obtained.

On average, there were 344.42 SNP markers mapped to each chromosome. This ranged from a low of 182 SNP markers on chromosome 12 to a high of 562 SNP markers on chromosome 1. With an average of 505.26 cM per chromosome, the total length of the genetic linkage map was 6063.12 cM (ranging from 343.72 cM on chromosome 10 to 676.52 cM on chromosome 2). From 0.92 cM on chromosome 6 to 2.89 cM on chromosome 12, the average genetic distance between pairs of SNP markers was calculated to be 1.58 cM. One single nucleotide polymorphism (SNP) marker was located approximately every 1.5 centimorgans (cM). This high-density genetic linkage map covered 373 Mb of the rice genome, allowing for more precise and reliable identification of QTLs for use in rice breeding environments experiencing DS. As a result of the high-density genetic linkage map, candidate genes inside the QTL areas that contribute to enhanced drought resistance in rice plants were identified and selected, and new drought-resistant rice cultivars can be developed. High-density linkage maps have been utilized in numerous earlier studies for QTL mapping [49,50,51,52,53,54].

### 2.8. QTL Mapping of Morphological and Yield Traits under Reproductive-Stage Drought Stress Conditions and Root Architectural Traits under ABA Conditions

In this study, 41 QTLs for root architectural traits under ABA treatments, in addition to morphological traits and grain yield components under WW and DS treatments, were identified (Appendix A, Figure 6). The detected QTLs changed in response to the WW, DS, and ABA treatments. For morphological features, grain yield components, and root architecture traits, the quantity of QTLs also varied. For morphological traits, a total of 3 QTLs, including *qPHC1.1*, *qPHC2.1*, and *qPHC2.2*, were discovered. Moreover, 23 QTLs were detected for grain yield component traits, such as *qTPBC4.2*, *qTPBC6.2*, *qTPBC10.1*, *qTPBD3.1*, *qTPBD9.1*, *qTPBD10.2*, *qTPBD10.3*, *qTPBD12.2*, *qPPBC4.1*, *qPPBC7.1*, *qPPBC8.1*, *qPLC7.1*, *qPLD4.2*, *qSPC7.1*, *qSPC7.2*, *qSPC8.1*, *qSPD7.1*, *qSPD10.1*, *qFGC5.1*, *qFGC6.2*, *qFGC7.1*, *qFGD5.1*, and *qFGD8.1*. For root architectural traits, 15 QTLs were identified, including *qRLA3-2.3*, *qRLA3-3.1*, *qRLA3-4.2*, *qRLA3-5.1*, *qRLA3-6.1*, *qRLA3-8.1*, *qRLA3-10.1*, *qRLA3-10.3*, *qRLA5-2.2*, *qRLA5-3.1*, *qRLA5-4.2*, *qRLA5-5.1*, *qRLA5-6.1*, *qRLA5-8.1*, and *qRLA59.1*. For both grain yield components and root architecture features, 2–4 QTLs were found to be co-localized in a total of 12 QTL clusters or hot spots, suggesting that these QTLs have the pleiotropic effect of a single gene or multiple connected genes [55]. In cluster 1, *qRLA3-2.3* and *qRLA5-2.2* are located on chromosome 2; in cluster 2, *qRLA3-3.1*, *qRLA5-3.1*, and *qTPBD3.1* are on chromosome 3; in cluster 3, *qRLA3-4.2*, *qRLA5-4.2*, *qPLD4.2*, and *qTPBC4.2* are on chromosome 4; in cluster 4, on chromosome 5. The significant positive association between grain yield components under DS and root architectural features under ABA conditions (Appendix A) suggests that QTLs for both traits were co-localized in the same or closely linked chromosomal regions. To exclude irrelevant candidate regions, our research only considered LOD values greater than 2.5 (Appendix A). The highest values of LOD were 28.8 and PVE was 13.81%.

### 2.9. Candidate Genes Underlying QTL Regions

As a result of our study, we were able to identify candidate genes that are involved in a wide variety of biological processes, molecular functions, cell components, and the response to drought (Appendix A). Out of 161 total QTLs, 41 QTLs with high LOD and PVE scores were further studied. Each QTL cluster is associated with one or more traits, suggesting the presence of putative genes in each QTL regions that have substantial pleiotropic effects. The positions of the peak markers for the QTL regions were converted to gene positions in base pairs and were annotated based on the Rice Genome Annotation (Osa1) Release 7. Each QTL peak was further analyzed, and a number of annotated and novel genes spanning that region within a 25 kb frame were chosen to determine the potential candidate genes associated with abiotic stress and maintenance of biological processes under both control and drought conditions (Appendix A). The two QTL regions (*qPHC2.1* and *qTPBD10.3*) with the highest LOD and PVE scores among the 41 QTL regions were examined for changes in gene expression levels under control and drought conditions in the parents (KB and ZHE733).

The *qPHC2.1* region has a LOD score of 24.42 and a PVE of 13.81% for the trait PHC. This *qPHC2.1* window region of 25 Kb upstream and downstream was chosen based on the location of the polymorphic SNP on chromosome 2. This region contains six genes with unknown annotations: LOC_Os02g44590, LOC_Os02g44599, LOC_Os02g44610, LOC_Os02g44620, LOC_Os02g44630, and LOC_Os02g44642 (Appendix A). The region *qTPBD10.3* also had a high LOD score of 21.82 and PVE of 8.49% and is associated with traits TPBD. The region spanning the polymorphic marker selected included five genes in chromosome 10: LOC_Os10g07030, LOC_Os10g07040, LOC_Os10g07050, LOC_Os10g07060, and LOC_Os10g07080 with an unknown annotation. The region (*qPHC1.1*) has nine genes spanning the 25 kb region and one of gene LOC_Os01g66270 (*qPHC1.1*, ERF/ethylene response factor) is associated with abiotic stress [56,57,58,59]. The two QTL regions *qRLA3-2.3* and *qRLA5-2.2* had common overlapping regions comprising five genes; one was the gene LOC_Os02g54160, with a APETALA2/ERF transcription factor, which has been studied for abiotic stress [60,61,62]. The QTL region with multiple traits, *qTPBD3.1*, *qRLA3-3.1*, and *qRLA5-3.1*, has six common genes, with one gene LOC_Os03g56280 annotated as a malate dehydrogenase [63]. The regions *qTPBC4.2*, *qRLA3-4.2*, and *qRLA5-4.2* were identified and shared the same region that had seven genes with one gene, LOC_Os04g44190 annotated for light reaction photosystem II [64]. The overlapping region of the *qFGC5.1*, *qFGD5.1*, *qRLA3-5.1*, *qTPBC6.2*, *qFGC6.2*, *qRLA3-6.1*, and *qRLA5-6.1* had a majority of 12 genes, 3 of which are linked to the response to drought-WRKY transcription factor (LOC_Os05g49210) [36,65], MYB protein (LOC_Os05g49240) [66,67,68] and Brassinosteroid/BR signaling (LOC_Os06g4908) [69,70,71,72]. The QTL region *qSPD7.1* contains nine genes and one potential candidate gene, LOC_Os07g02520, which regulates sugar partitioning in carbon-demanding young leaves and developing leaf sheaths [18]. The regions *qFGD8.1*, *qRLA3-8.1*, and *qRLA5-8.1* contain six genes, one of which is involved in stress, LOC_Os08g06344 [73,74,75]. The regions *qTPBD9.1* and *qRLA59.1* contains 7 genes, 1 of which is involved in carbohydrate metabolism (LOC_Os09g08120) [73] and *qTPBD12.2* has 10 genes and LOC_Os12g02700 is annoted as a late embryogenesis abundant/LEA protein [76,77,78,79,80,81].

### 2.10. RT-qPCR Validation of the Key Functional Genes Identified within the QTL Regions Regulating Drought-Related Traits and ABA Sensitivity

The 26 potential candidate genes identified from multiple QTL regions that had high LOD and PVE scores were selected for gene expression variations in parents KB and ZHE733 under control and drought conditions. Amongst them, 15 candidate genes have been previously studied for abiotic stress and 11 are novel candidate genes. The QTL regions that were up-regulated under control and drought for KB parent compared to ZHE733 were LOC_Os02g54160 (*qRLA3-2.3*, *qRLA5-2.2*), LOC_Os03g56280 (*qTPBD3.1*, *qRLA3-3.1*, *qRLA5-3.1*), LOC_Os08g06344 (*qFGD8.1*, *qRLA3-8.1*, *qRLA5-8.1*), LOC_Os10g41460 (*qRFWA3-10.1*), LOC_Os11g30760 (*qRFWA3-11.1*), LOC_Os12g29330 (*qTRN12.1*), LOC_Os02g44610 (*qPHC2.1*), LOC_Os10g07040 (*qTPBD10.2*), and LOC_Os10g07080 (*qTPBD10.2*) suggesting that these genes are constitutively highly expressed in control conditions and are also maintained with higher expression in drought conditions in parent KB compared to ZHE733.

The gene in the QTL region *qPHC1.1* (LOC_Os01g66270) was down-regulated in KB under control conditions compared to ZHE733, but under drought conditions, this gene shows almost 3.5 log2FC up-regulation in KB compared to ZHE733. The following genes in the QTL region, LOC_Os07g02520 (*qSPD7.1*), LOC_Os09g08120 (*qTPBD9.1*, *qRLA59.1*), LOC_Os12g02700 (*qTPBD12.2*), LOC_Os02g44620 (*qPHC2.1*), LOC_Os02g44630 (*qPHC2.1*), and LOC_Os02g44642 (*qPHC2.1*) are up-regulated under control conditions but down-regulated in KB compared to ZHE733 under drought.

From gene expression analysis, we can conclude that in the region *qTPBD10.3* with high LOD and PVE scores, the two identified novel genes, LOC_Os10g07040 and LOC_Os10g07080, could be potential candidates for total plant biomass under drought based on gene expression levels in control and drought conditions. Among the candidate drought-resistance genes not annotated to drought stress functions, LOC_Os10g07040 showed high up-regulation in Kaybonnet compared to ZHE733 under DS and WW conditions (Figure 7A,B), correlated with chalcone synthase, according to the MSU rice reference genome annotation release 7.0 that is involved in the drought stress response in rice [82], Arabidopsis [83], and tobacco [84]. The other candidate drought-resistance gene that also showed to be highly up-regulated in Kaybonnet compared to ZHE733 under DS and WW conditions (Figure 7A,B) was LOC_Os10g07080, which is related to myosin [85] and transposon proteins [86] that regulate cell growth and developmental processes in rice. These candidate genes could be functioning in a cumulative manner in order to show a measurable positive effect on improving drought resistance in rice, and the effect of genes can further be exploited to develop drought-resistant cultivar.

### 2.11. Genetic Diversity in 26 Loci across K/Z RIL Population

The 26 drought loci described earlier are distributed across all 12 chromosomes with 2 distinct clusters on chromosome 2 (6 loci) and chromosome 10 (5 loci). The up- and down-stream region of the 26 drought loci was used for dissecting the genetic diversity in and surrounding these loci in the diversity panel. Depending upon the proximity to the nearest neighbor gene, the region of interest was defined between 400 bp to 22 Kbp for the upstream and 800 bp to 18 Kbp for the downstream region.

We identified a total of 7475 SNPs across these 26 loci in the 200-genotype of the K/Z RIL population. The diversity panel represents six distinct subpopulations (Ind, *indica*; Aus, *aus*; TeJ, *temperate japonica*; TrJ, *tropical japonica*; Adm, *admixture;* and Aro, *aromatic*). Each individual SNP has multiple effects on individual loci depending upon the ‘impact’, ‘functional class’, ‘type’, and ‘region’ where it is present as shown (Appendix A). The SNPs were predominantly found in the upstream/promoter and intergenic regions. Five SNPs resulted in a stop codon gained in three loci (one stop gained in LOC_Os09g08120.1 and two stops gained in LOC_Os10g07030.1 and LOC_Os10g07050.1 each). SNP annotation showed that although most variants are moderate/modifying in effect, there are 7 high-impact and 200 low-impact SNP effects that are potential candidates for further testing and validation. Moreover, the two QTLs on chromosomes 2 and 10 containing 6 and 5 loci, respectively, show a distinct pattern of variation as compared to the overall variation in 26 loci.

To determine how the variations present in these 26 loci differentiates the 200 genotypes, we performed a principal component analysis (PCA) on the vcf files containing the SNP datasets. We performed three separate runs of PCA for (i) all 26 loci, (ii) 6-loci cluster on chromosome 2, and (iii) 5-loci cluster on chromosome 10 containing 7475 SNPs, 565 SNPs, and 4375 SNPs, respectively (Appendix A). In the PCA for 26 loci, the first principal component (PC1) explains 34% of the variation in the data, and clearly separates a subset of *indica* subpopulation from the rest of the genotypes (Appendix A). This is in contrast to the PCA performed for 6.5 million SNPs representing the whole genome, where the *indica* subpopulation segregates as a single group. SNPs on chromosome 10 cluster also show a similar pattern of separation of *indica* subpopulations into two distinct groups with PC1 explaining 49% of the variation in the data (Appendix A). However, the SNPs on chromosome 2 cluster show an interesting pattern where PC1 clearly separates the *japonicas* (both temperate and tropical) from the *indica* and *aus* subpopulations, and explains 84% of the variation in the data in doing so (Appendix A). This indicates the presence of novel variation in the six drought loci on chromosome 2 (LOC_Os02g44590, LOC_Os02g44599, LOC_Os02g44610, LOC_Os02g44620, LOC_Os02g44630, LOC_Os02g44642) that distinguish the *japonicas* from the *indica* subpopulations.

## 3. Discussion

The K/Z RIL population was developed at the USDA Dale Bumpers National Rice Research Center in Stuttgart, AR, USA. This was accomplished by crossing diverse parental genotypes from different subspecies, namely Kaybonnet (*tropical japonica*) and ZHE733 (*indica*), using the SSD method. This was performed in order to create segregating progenies with high genetic variability for selection-desirable genes. The many genotypes of rice investigated here exhibit a wide variety of patterns of root growth and development. Greub [87] observed that when compared to other rice genotypes, such as Bengal, Sipirasikkam (GSOR 310428), O. glaberrima, IR64, Nagina-22 (N22), and Vandana, the root length of Kaybonnet showed the longest root. Because it enables the roots to reach deeper water levels in the ground, root length is likely the single most important architectural characteristic for protecting against drought. In addition, Greub [87] found that the number of metaxylems in the Kaybonnet variety was larger than in Bengal, *O. glaberrima*, IR64, Nagina-22 (N22), Vandana, and the parental line ZHE733. There is a correlation between the number of metaxylem vessels and their size and the water conductivity of the tissue. Among these lines, Bengal and Nagina-22 (N22) are examples of drought-resistant rice genotypes. On the other hand, IR64 and Nippobare serve as references for drought-sensitive rice genotypes.

According to Huang et al. [88], as compared to other cereal crops including wheat, rye, and barley, the susceptibility of rice to DS conditions is significantly higher. The plant height (PH), tiller number (TN), heading day (HD), flag leaf width (FLW), and leaf rolling score (DLR) are all different between the two parental lines. Other morphological characteristics include heading day (HD), flag leaf width (FLW), and rolling score (DLR). In comparison to ZHE733, the donor parent Kaybonnet has characteristics such as taller stature, lower TN, later HD, wider FLW, and lower DLR, whereas ZHE733, the recurrent parent, has characteristics such as shorter stature, higher TN, earlier HD, narrower FLW, and higher DLR under WW conditions. Kaybonnet was developed from the varieties Katty and Newbonnet. Kaybonnet showed overall higher performance versus ZHE733 due to the fact that Kaybonnet is drought-resistant, whereas ZHE733 is susceptible to DS conditions, hence showing a lower resistance. This is in addition to the fact that the two parental lines exhibited variation for morphological features when they were treated to DS conditions. The PH of Kaybonnet decreased to 14.29% whereas that of ZHE733 dropped to 39.22% when the circumstances were DS. Although, the TN of Kaybonnet declined to 20%, and ZHE733 decreased to 14.3%. Previous research has shown that water deficit conditions have a negative effect on plant development and growth due to a loss of turgor [89,90,91,92,93]. These results are in consensus with that research since they show that water deficit conditions have a negative effect on plant development and growth. According to Mantovani and Iglesias [94], the reduction in plant height and the quantity of productive tillers in DS circumstances are associated with a reduction in the cell cycle processes, including cell expansion and elongation.

All morphological traits within the RIL population under DS were significantly lower than under WW conditions (Appendix A). In WW conditions, the RIL population is more skewed toward the parent ZHE733 in terms of PH and TN characteristics with plants taller than 50 cm and having more than 8 tillers. Under DS conditions, the average PH in the RIL population decreased by 50%, which is more than either parent, but the average TN decreased by 13.90%, which is less than either parent. Based on the PH reduction in the DS conditions relative to the WW conditions, 54.04% of lines in the RIL population showed a reduction greater than 50%, indicating that more than 50% of the lines are sensitive to DS conditions. Meanwhile, based on the TN, 71.72% of lines in the RIL population displayed drought resistance with a reduction of less than 30%, indicating that the majority of lines have reduced the tiller number and exhibit ZHE733 traits. In the RIL population, 28 lines indicated a reduction of 30% or less for both plant height and tiller number; these lines represent a potential genetic resource for developing drought-resistant varieties that are less affected by drought stress. According to our research, plant height and tiller number are still affected by scarcity during the reproductive stage. After the reproductive stage, the rice plants continued to grow and had been in the tilling stage. Several studies, including Islam et al. [95], Manickavelu et al. [96], Sarvestani et al. [97], Mostajeran and Rahimi-Echi [98], Ji et al. [99], Ashfaq et al. [100], and Sokoto and Muhammad [101], supported these findings.

At each growth stage, DS conditions have distinct impacts on rice. Rice is extremely sensitive to DS conditions, particularly during the reproductive phase, and cereal yield is drastically reduced even under mild DS conditions [102,103,104]. Developing drought-resistant rice varieties is the most effective method to reduce grain yield loss under DS conditions; however, this is extremely difficult due to the complexity of the drought-resistance trait associated with grain yield components. The identification of drought QTLs associated with crop yield components under water deficit conditions and their use in molecular breeding is an alternative strategy for enhancing the breeding efficiency. Additionally, QTLs can be incorporated into a marker-assisted breeding (MAB) strategy [105]. The mean values of the grain yield components, including total plant biomass (TPB), spikelet per panicle number (SP), filled grain per panicle number (FG), panicle length (PL), and primary panicle branch number (PPB), differed significantly between DS and WW conditions for both parents and the population. ZHE733 exhibited a larger decrease in total grain yield components than Kaybonnet. These RIL lines are a potential genetic resource for developing drought-resistant rice varieties that are more resistant to the reproductive-stage drought stress condition. Twelve lines exhibited less reduction (30%) for all grain yield components.

The parents, Kaybonnet and ZHE733, as well as the RIL population, exhibited a range of responses to ABA conditions, with Kaybonnet, the drought-resistant parent, exhibiting greater ABA sensitivity than ZHE733, the drought-sensitive parent. Both parents’ RL, TRN, SRN, DRN, and RFW decreased under ABA conditions compared to control conditions. Kaybonnet demonstrated a greater decrease in RL, TRN, SRN, DRN, and RFW than ZHE733. Lim et al. [106], Duan et al. [107], and Todorov et al. [108] have shown that ABA sensitivity is associated with drought stress resistance through its impact on stomatal movement. In addition, Lim et al. [106] found that drought-sensitive rice plants grown in media containing 2 uM or 5 uM ABA had substantially longer roots and shoots than plants grown in control media. Based on these findings, the data indicates that the rice plants that were sensitive to drought were not affected by ABA and that their response to drought stress followed an ABA-dependent pathway.

In this study, a positive correlation was found between FG-DS and the majority of the morphological traits, the other grain yield components, and the major root architectural traits under ABA conditions. This indicates that the rice drought-resistant plants maintain their grain yields under DS conditions through the development of cell elongation, the maintenance of cellular membrane integrity, and the regulation of osmotic stress tolerance via ABA-mediated cell signaling [109]. In addition, the fact that the FG-DS has a negative association with TPB under both WW and DS conditions suggests that there is a greater assimilate distribution in the grains than in the vegetative components of the plant. In addition, it was discovered that the sensitivity of rice plants to ABA was connected with their ability to withstand drought [106,107,108].

Many studies have been reported that various root anatomical phenes such as metaxylem, aerenchyma, and cortical cells properties influence plant performance and productivity under DS conditions in cereal crops including maize [110,111,112,113], rice, and wheat [114], and also in legume crops such as soybeans [115], common beans [116], chickpeas, groundnuts, pigeonpeas, and cowpeas [117]. Under DS conditions, plants allocate more energy and carbon resources to root growth rather than to shoot growth, which can increase water acquisition [118,119,120]. Aerenchyma and cortical cells contribute to increased root growth for soil exploration in the deeper soil layer and, consequently, greater water acquisition and improved productivity by reducing the root metabolic costs, thus leading more carbon resources to be allocated to improve root growth [121,122]. Understanding root adaptive mechanisms is important to maintain higher productivity under DS conditions.

The movement of water and nutrients from the soil into the plant cells is facilitated by the metaxylem vessels [123]. Metaxylem vessel traits are significant characteristics to improve water-use efficiency under DS conditions according to a study by Kadam et al. [114]. In comparison to ZHE733 and drought-sensitive lines, parental Kaybonnet and drought-resistant lines displayed a higher number of metaxylem vessels. Kaybonnet has two metaxylem vessels, while drought-resistant lines typically have four, compared to just one in ZHE733 and drought-sensitive lines. Under DS conditions, an increase in the number of metaxylems causes an increase in root hydraulic conductivity, which lowers the metabolic costs of exploring water in deeper soil layers and increases water uptake efficiency and yield [115]. Under DS conditions, the improved root hydraulic conductivity also results in improved performance of the shoot physiological processes. The more metaxylem vessels there are, the more effectively water is absorbed under DS conditions, which increases stomatal conductance, internal carbon dioxide capture, and overall maintains photosynthetic activity. In contrast, metaxylem size was smaller in drought-resistant lines than it was in drought-sensitive lines. Drought-resistant lines only have a metaxylem size of 0.0005 mm^2^, while drought-sensitive lines have an average metaxylem size of 0.0008 mm^2^. Reduced metaxylem size increases yield under DS conditions because it is connected with higher hydraulic conductivity and thus more efficient water collection [124].

Aerenchyma, which impacts root respiration, is the increased air gap in the root cortex as a result of programmed cell death [125]. In rice, there was a significant connection between aerenchyma and drought resilience. In comparison to ZHE733 and drought-sensitive lines, Kaybonnet and drought-resistant lines displayed higher percentages of aerenchyma. Higher aerenchyma is linked to lower root respiration costs, which results in more carbon being allocated and better root growth, thus increasing water uptake from the deep soil layer of drying soil, and higher grain yields under DS conditions [42].

The presence of cortical cells in the roots is another anatomical phene critical for enhanced drought resistance. Total transverse root cortex minus aerenchyma area equals cortical cell area. Drought-sensitive lines have an average of 11 cortical cell files, while drought-resistant lines have just 5. Drought-sensitive lines have higher respiration compared to drought-resistant lines, leading to reduced root growth, lower water acquisition, and decreased productivity under DS conditions [121,122]. The number of cortical cell files was found to be positively correlated with root hydraulic conductivity and water uptake under DS conditions [126].

To comprehend the genetic complexity of the drought-related features, it is crucial to identify QTLs for morphological traits, grain yield components, and root architecture traits. Numerous genes that have a significant impact on the traits control the genetic variation of traits connected to drought [127]. For morphological features and grain yield components under the WW and DS treatments, as well as root architecture traits under the ABA treatments, 41 QTLs were found in this study. For morphological features, grain yield components, and root architecture traits, the quantity of QTLs also varied.

The majority of the QTL found in this study were mapped to regions that were similar to those found in earlier studies [16,128,129,130,131,132,133,134]. According to many studies (including those by Lanceras et al. in 2004 [130], Zhou et al. in 2016 [135], Jiang-xu et al. in 2016 [136], Yadav et al. in 2019 [137], Zeng et al. in 2019 [138], and Xu et al. in 2020 [139]), chromosome 1 has the highest concentration of QTLs for PH (*qPHC1.1*). Most of the QTLs connected to grain yields were found on chromosomes 5 and 6 (*qTPBC6.2*, *qFGC5.1*, *qFGD5.1*, and *qFGC6.2*) [45,127,131,140,141,142]. Additionally, according to Xu et al. [142], Lou et al. [143], Kitomi et al. [144], and Gimhani et al. [145], chromosome 10 had the most QTLs (*qRLA3-10.1* and *qRLA3-10.3*) for root architectural traits.

Identification of candidate genes within a QTL region is essential for developing transgenic rice with enhanced drought resistance [146]. Within the 41 QTL regions for morphological traits, grain yield components under WW and DS conditions, and root architectural traits under ABA conditions, 184 candidate genes linked in numerous biological processes, molecular functions, cell components, and drought response were discovered. Approximately 15 candidate genes with a high LOD and PVE score that have been studied for drought stress previously. In this analysis, 11 novel candidate genes with high LOD and PVE scores and unknown annotations were identified. Within these QTL regions are numerous known genes with homology to APETALA2 (AP2)/ethylene response factor (ERF) transcription factor, malate dehydrogenase protein, photosystem II oxygen evolving complex protein PsbQ family protein, WRKY transcription factor, MYB transcription factor, zinc finger (ZFN) protein, endoplasmic reticulum protein, DEAD-box RNA helicase, glycosyl transferase, late embryogenesis abundant (LEA) protein, and no apical meristem protein (NAC).

A transcription factor family well-known for drought response such as APETALA2 (AP2)/ethylene response factor (ERF) [56,57,58,59] has family members present in several of the QTL regions such as QTLs for PH-WW on chromosome 1 (LOC_Os01g66270); RL-ABA3 and RL-ABA5 on chromosome 2 (LOC_Os02g54160); and FG-WW, FG-DS, RL-ABA3, and RL-ABA5 on chromosome 5 (LOC_Os05g49010). Moreover, the chromosome 1 QTL regions for PH-WW are adjacent to the semi-dwarfing gene sd1 locus (38.3 Mb). Vikram et al. [147] discovered a robust association between *sd1* and QTL for drought-related traits. The *sd1* locus is also linked to above- and below-ground traits in rice, such as plant height and root architectural traits [148]. The adaptation of rice plants to DS conditions is a reduction in plant height. A significant QTL for grain yield components under DS, FG, which is located on chromosome 5 overlaps with 12 candidate genes. In addition, there is an overlap between QTL for root architectural traits, RL under ABA conditions and FG under DS, indicating that ABA is implicated in the mechanism for drought stress resistance. Under DS conditions, ethylene biosynthesis is enhanced and interacts with AP2/ERF, resulting in a water deficit response [60,61,62]. AP2/ERF also responds to ABA to help activate ABA-dependent and -independent stress-responsive genes. Pan et al. [149] observed an increase in drought resistance in transgenic rice that overexpressed AP2/ERF. Understanding the functions of the AP2/ERF gene in the mechanisms of rice’s drought resistance could provide valuable information for enhancing the plant’s ability to adapt to DS conditions.

LOC_Os03g56280, a significant candidate gene known to modulate malate dehydrogenase in response to drought stress, was identified in the QTL regions for TPB-DS, RL-ABA3, and RL-ABA5, on chromosome 3. In the QTL regions for TPB-DS and RL-ABA5, a candidate gene for carbohydrate metabolism was identified on chromosome 9 (LOC_Os09g08120). Malate dehydrogenase is an enzyme that uses NAD(H)/NADP(H) as a cofactor to catalyze the oxidation of malate to oxaloacetate. In addition, this enzyme can be expressed in various rice plant tissues, including the root, leaf, panicle, and stem, and was induced by water deficit [63]. Transgenic plants that overexpress malate dehydrogenase are more resistant to drought than their wild-type counterparts. Malate dehydrogenase was also identified by Agrawal et al. [150] as a drought-responsive protein [151]. Drought-resistant genotypes accumulate a greater amount of malate dehydrogenase, which protects membranes from harm by reactive oxygen species (ROS) under DS conditions.

A photosynthesis-encoding gene (LOC_Os04g44190) is present in the TPB-WW, RL-ABA3, and RL-ABA5 QTL regions on chromosome 4. This gene regulates stomatal closure and protects plants from dehydration [64]. It is implicated in the light reaction of photosystem II (PSII). Due to a decrease in photosynthetic electron transport and carbon assimilation, the photosynthetic rate is reduced under DS conditions, resulting in a decrease in grain yield. In addition, PSII is a pigment-protein complex in thylakoid membranes responsible for oxygen evolution, water splitting, and plastoquinone reduction [152]. LOC_Os04g44190 encodes a PsbQ family protein that is a member of the class of PSII extrinsic proteins; under DS conditions, the expression of this protein is altered owing to a change in PSII efficiency [64]. Consequently, the PsbQ protein serves a crucial role in drought resistance.

On chromosome 5 (LOC_Os05g49210) of the QTL regions for FGC and FGD, a WRKY transcription factor involved in drought stress response and plant development was identified. Shen et al. [153] reported that transgenic rice with increased OsWRKY30 expression exhibited enhanced drought resistance. Similarly, silencing WRKY genes in transgenic rice increased its sensitivity to drought. In addition, expression of the WRKY transcription factor under DS conditions induced ABA accumulation, resulting in stomatal closure and a reduction in water loss [36,65]. Yan et al. [154] reported that ABA treatment increased the expression of the WRKY transcription factor.

On chromosome 5 (LOC_Os05g49240) in the QTL regions for SP-DS and PPB-DS; and also on chromosome 10 (LOC_Os10g41460) in the QTL regions for TPB-DS and RL-ABA3, were identified MYB transcription factor, a well-known transcription factor in drought response [66,67,68]. Rice transgenic with an overexpression of OsMYB6 was less susceptible to drought than its wild-type counterpart and had increased proline catalase (CAT) and superoxide dismutase (SOD) activity. In addition, under DS conditions, OsMYB6 transgenic rice plants exhibited increased expression of abiotic stress-responsive genes [66]. The expression of MYB genes is regulated by drought, according to Katiyar et al. [155]. Xiong et al. [156] found a correlation between the overexpression of MYB genes and ABA accumulation and increased drought resistance.

Zinc finger (ZFN) protein (LOC_Os06g49080), a stress-responsive transcription factor, is located on chromosome 6 in the QTL regions for FG-WW, BY-WW, RL-ABA3, and RL-ABA5. It has been reported that the ZFN protein enhances drought resistance in plants, suggesting that the ZFN protein contributes to the higher yield under DS conditions by regulating stomatal closure [69,70,71,72]. OsSAC1, a gene involved in sugar metabolism, was identified on chromosome 7 (LOC_Os07g02520) in the SP-DS QTL regions [25]. OsSAC1 regulates sugar partitioning in the carbon metabolism of juvenile leaves and developing leaf sheaths. OsSAC1 encodes an endoplasmic reticulum protein that is responsible for sucrose accumulation in rice leaves and can be used to generate energy and build carbon skeletons.

The genomic region on chromosome 8 (LOC_Os08g06344) within the QTLs for FG-DS, RL-ABA3, and RL-ABA5 encodes a DEAD-box RNA helicase that has been reported to enhance drought resistance in rice [66,67,68]. Over-expression of OsRH58, a chloroplast DEAD-box RNA helicase, in transgenic rice improved drought resistance as measured by a higher survival rate under DS conditions compared to the wild type [73]. Furthermore, under drought conditions, OsRH58 gene expression was elevated.

A gene regulating late embryogenesis abundant (LEA) protein was identified on chromosome 12 (LOC_Os12g02700) between the TPB-DS QTL regions. The LEA protein is essential for drought resistance in plants [76,77,78,79,80,81]. Under DS conditions, drought-resistant plants expressed more LEA genes than drought-sensitive plants. In support of LEA functions, Xiao et al. [76] reported that transgenic rice with over-expression of the LEA protein gene OsLEA3-1 produced a higher grain yield than wild-type rice under DS conditions.

Within the QTL region for SP-DS on chromosome 12, LOC_Os12g29330 (OsNAC139) was identified. OsNAC139 is a member of the NAC transcription factor family that is known to control plant response to drought [157,158,159,160] by producing no apical meristem (NAM)/NAC protein. Rice contains 151 NAC genes [161], from which several studies have reported that over-expression of OsNAC genes leads to improved drought resistance [158,162,163,164,165].

Among all the candidate genes identified within the QTL regions, various transcriptomes correlated with drought stress resistance were detected. Drought resistance of the rice plants can be either associated with metabolic regulation or osmoregulation. This study has revealed a number of potential candidate genes for developing drought-resistant rice varieties. Information about the differences in the expression of drought-resistance genes between drought-resistant and sensitive genotypes and their relationship to morphological characteristics, grain yield components, and root architectural traits under DS conditions has been limited. However, the study of the differences in gene expression between resistance and sensitive genotypes could improve the efficiency and opportunities of developing drought-resistant varieties.

A large number of genes were up-regulated in Kaybonnet (drought-resistant parent), indicating that the drought-resistant cultivar had a higher capability to modulate drought-resistant genes when exposed to DS conditions, thereby enhancing its resistance level compared to drought-sensitive parent (ZHE733). Modulation of a higher number of up-regulated expressed genes with different transcription factor gene families is a crucial characteristic of drought-resistant genotypes. Similar results were also obtained by Hayano-Kanashiro et al. [166] who showed that drought-resistant maize genotypes inducted more genes compared to the sensitive genotypes under DS conditions. All 15-candidate drought-resistance genes identified within QTL regions have been strongly associated with direct roles in drought stress resistance. For example, transcription factors MYB, NAP, NAC, ZIP, and APETALA2/ERF are responsive to dehydration induced by water deficit conditions [56,57,58,59,66,67,68,156,158,162,163,164,165,167,168,169,170]. These results provide strong evidence for genes expressed under DS conditions being involved in various physiological, biochemical, and molecular processes within the rice, in order to reduce the effects of drought stress, thereby enhancing their ability to resist the drought stress and maintain their grain yield production under DS conditions. Therefore, the up-regulation of the drought genes in Kaybonnet compared to ZHE733 provides important information to characterize the function of candidate drought-resistance genes and to understand the drought stress mechanisms in rice.

Candidate genes within QTL regions involved in regulatory response to drought include a large family of genes expressed under DS conditions. Proteins expressed by known and candidate drought-resistance genes played important roles in (1) cellular protection, including structural adaptation and osmotic adjustment, and (2) drought responses by interaction with other proteins and transcription factors, such as MYB, NAP, NAC, bZIP, and APETALA2/ERF. Under DS conditions, in the drought-resistant genotype Kaybonnet, exogenous ABA significantly improved the expression of ABA biosynthetic genes suggesting the Kaybonnet genotype must be maintaining the water potential and cellular activity of the cell by closing the stomata.

Based on the RT-qPCR results, it may also be suggested that there is a correlation between gene expression, transcriptional regulation, and resistance to drought across resistance and sensitive genotypes. Therefore, the up-regulation of the drought genes and novel candidate genes in Kaybonnet compared to ZHE733 provided important information to characterize the function of candidate drought-resistance genes. All in all, these results enhance our understanding of the role of candidate drought-resistance genes in the regulation of drought stress response, and this research has also revealed a number of potential candidate drought-resistance genes that could be used to develop rice cultivars with greater drought resistance. Additional study is needed to alter the gene sequences of locally grown rice cultivars to improve their drought tolerance when exposed to abiotic stress.

## 4. Materials and Methods

### 4.1. Plant Materials

A total of 198 RILs from an F2 population was generated by single seed descent (SSD) from the cross between varieties Kaybonnet (*Oryza sativa*, an upland *japonica* type) and ZHE733 (*Oryza sativa*, an *indica* type), K/Z RILs (USDA, Stuttgart, AR, USA) after selfing for F10 generations. An extensive study was performed for various morphological traits, such as grain yield components under well-watered (WW) and drought stress (DS) conditions, and also the response of ABA on root architectural traits.

### 4.2. Drought Stress Treatment at the Reproductive Stage

To synchronize the drought treatment at the reproductive stage, seeds of the K/Z RIL population and both of its parents (Kaybonnet and ZHE733) were germinated, grown in the greenhouse, and uniform plants were transplanted to the field separately in 6 batches (at 7-day intervals). This RIL population was grown in the field in Fayetteville, Arkansas, during the growth seasons (May to November) in 2016, 2017, and 2018, with an average yearly rainfall of 849.63 mm. A randomized full-block design was implemented to grow the population, with five replications and two treatments—well-watered (WW) and drought stress (DS) conditions. The DS treatment was imposed at reproductive stage (R3). Using a tensiometer, the DS conditions were maintained continuously below −70 kPa (severe stress), and the WW plot was continuously watered.

A number of morphological traits and grain yield indicators, including heading day (HD), plant height (PH), productive tiller number (TN), flag leaf width (FLW), leaf rolling score (LR), spikelet per panicle number (SP), panicle length (PL), primary panicle branch number (PPB), filled grain per panicle number (FG), hundred grain weight (HGW), and total plant biomass (TPB) with five replications per line, were measured to determine the impact of drought stress.

Heading date was recorded as the period from the germinating date to the time when 50% of the panicles have exerted. Plant height was measured from the ground surface to the tallest panicle tip before harvesting with a ruler. The total of the productive tiller number per plant was recorded after 10 days of DS treatment. Productive tiller number means number of tillers bearing panicle at the reproductive stage. Flag leaf width was measured in the widest area of the leaf on the tallest culm of each plant by using a ruler after 10 days of the DS treatment. The leaf rolling score on the first five leaves on the tallest tiller of each plant was identified after 10 days of DS treatment based on the standard evaluation system for rice [171]. The range in score is from 1 to 5, 1 indicating unrolled leaves and fully turgid, 2 indicating leaves are folded (deep-V-shaped), 3 indicating leaves are fully cupped (U-shaped), 4 indicating leaves margins touching (O-shaped), and 5 indicating completely rolled leaves.

The spikelet per panicle number was counted manually for each panicle. Panicle length, the length per panicle, was measured from the panicle neck to the panicle tip. Primary panicle branch number was determined by counting the branches that directly come out from the peduncle. Manual counting was used to determine filled grain per panicle number. The hundred grain weight was calculated based on the weight of one hundred filled grains of each plant with 14% grain moisture content. Total plant biomass was determined using the total above-ground parts of the rice plant (panicles, stems, and leaves) at the maturity stage, which were oven-dried at 80 °C for 72 h and then weighed.

Using JMP version 12.0, the data for morphological features and grain yield components under WW and DS conditions were evaluated using an analysis of variance and the Tukey’s HSD was used to compare the means of the two treatments (WW and DS) (Tukey’s HSD, P 0.05). Using the Shapiro–Wilk test, a normal distribution for each trait was found by using SAS 9.4.

### 4.3. Screening for ABA Sensitivity

The seeds from 198 recombinant inbred lines and a total of 2 parental genotypes were sterilized using 70% ethanol for 60 s and 30% bleach solution for 45 min, followed by four rinses with sterilized water. To ensure consistent seedling growth, sterilized seeds (S0 stage) were germinated in 2 mL tubes containing germination media (Chu’s N-6 basal salts with vitamins, macronutrients, and micronutrients) until S3 stage in a growth chamber (temperature: 28/22 °C day/night, light intensity: 600 umol/m^2^/s, relative humidity: 60%) for uniform seedling growth. The Kaybonnet, ZHE733, and 198 lines’ seedlings at the S3 stage were transplanted into ABA media at various concentrations—0 M (control), 3 M, and 5 M—and then cultivated in a growth chamber until the V3 stage. The root architectural traits of maximum root length (RL), root–shoot ratio (RSR), total root number (TRN), number of roots with a shallow angle (0–45°) (SRN), number of roots with a deep angle (45–90°) (DRN), and root fresh weight (RFW) were measured with five replications per line/treatment in order to quantify the effect of ABA sensitivity.

### 4.4. Measurements of Root Anatomical Phenes

The root anatomical phenes of the genotypes (Kaybonnet and ZHE733), 18 drought-resistant lines, and 18 drought-sensitive lines were measured with five biological replications using root samples from the V3 growth stage. A section of approximately 10 cm from the midsection of the primary root was used for fixing. For fixation, root segments were immersed in an FAA (formaldehyde alcohol acetic acid, 10%:50%:5% + 35% water) solution for at least 24 h at 4 °C, followed by dehydration in a series of ethanol concentrations of 70%, 80%, 85%, 95%, and 100% for one hour each. The dehydrated samples were then treated twice for one hour with toluene and then embedded in paraffin. The embedded samples were sectioned 80–250 m with a Microtome (Accu-Cut SRM 200 Rotary Microtome, Sakura Finetek USA, Inc., Torrance, CA, USA). The root sections were mounted on microscope slides and dried overnight at 37 °C in a forced air dryer. The root sections were stained for 3 min with 0.05% toluidine blue, de-stained with distilled water, air dried for 2 min, and then embedded in xylene. Images of root sections were captured using a digital camera (Nikon Optihot 2 and Nikon D1) mounted on a microscope with a 40× magnification. RootScan2 software (https://plantscience.psu.edu/research/labs/roots/publications/overviews/rootscan-software-for-high-throughput-analysis-of-root-anatomical-traits (accessed on 30 July 2023)) was used to measure the metaxylem, aerenchyma, and cortical cell properties of roots [172].

### 4.5. Genotyping

Genomic DNA was extracted from two-week-old seedlings of the 198 selected lines and parents (Kaybonnet and ZHE733) using the cetyl tri-methyl ammonium bromide (CTAB) approach. The GBS libraries and analysis were handled by the Genomics Center at the University of Minnesota. The enzymes *Pstl* and *Mspl* were used to generate a single-end library. One million base pairs (bp) of reads were produced by NextSeq for each sample. The average quality ratings of the selected reads were higher than Q30.

### 4.6. SNP Identification

Illumina BCL2FASTQ created de-multiplexed FASTQ sequencing readings to identify SNPs. FASTQ files with more than 2,000,000 readings were subsampled. Trimmomatic deleted adapter sequences at the 3′ ends of reads by removing the first 12 bases from each read. Burrows–Wheeler Alignment (BWA) linked the FASTQ data to the reference genome of Nipponbare, *Oryza sativa* spp. Japonica version MSU7. Aligned and mismatched nucleotides were used for SNP calling. All samples were concurrently collectively called for variations using Freebayes. All samples were collectively called variations using Freebayes. Freebayes’ raw variant call format (VCF) file was filtered using VCF tools to remove variants with minor allele frequency of less than 1%, variants with genotype rates less than 95%, samples with genotype rates less than 50%, variants with 100% missing data, variants with monomorphic markers between parents, and variants with more than 50% heterozygosity. The filtered data file containing the filtered SNPs in nucleotide-based hap map format was changed to an ABH-based format, where “A” represents donor allele, “B” represents recipient allele, and “H” represents heterozygous allele.

### 4.7. Linkage Map Construction and QTL Mapping

The QTL IciMapping software version 4.2.53 [173] linkage mapping function was used to build linkage maps for 198 lines of the K/Z RIL population with filtered SNP markers. The recombination frequency (r) was set at 0.45. The Kosambi mapping function converted recombination frequencies to map distance (cM) [174]. QTL analysis used morphological characteristics and grain yield components PH, SP, PL, PPB, FG, and TPB under WW and DS conditions. HD, TN, FLW, and DLR data were obtained. RL, RSR, TRN, SRN, DRN, and RFW were tested for ABA sensitivity at 0 μM (control), 3 μM, and 5 μM. Only RL was used for QTL mapping. QTL analysis was performed using 4133 SNP markers using QTL IciMapping version 4.2.53 with an inclusive composite interval mapping (ICIM) tool [173]. Significant QTLs for LOD ≥ 2.5 were identified. QTL nomenclature is determined by trait name, chromosome number, and genomic map position [175,176]. Left and right markers flanking QTLs were determined. The nearest marker to the QTL peak determined genotypic frequency.

### 4.8. Identification of Candidate Genes within the QTL Regions

Based on the position of the SNP markers bordering the QTL areas and the nearest predicted/annotated gene in the region, the MSU rice japonica reference genome annotation release 7.0 was used to identify potential genes. To identify the key functional genes regulating drought-related traits and ABA sensitivity, all genes within the sliding window of 25 Kb upstream and downstream of the identified QTLs were classified into the following three major functional categories: biological process, molecular function, and cellular component. RNA from the parental genotypes was extracted and used to analyze gene expression under DS conditions.

### 4.9. RT-qPCR Validation of the Key Functional Genes Identified within the QTL Regions Regulating Drought-Related Traits and ABA Sensitivity

The two parental genotypes’ leaves were collected for RNA extraction and gene expression measurement (Appendix A). The RNeasy^®^ Plant Mini Kit (Qiagen Inc., Hilden, Germany) with a 260/280 ratio of 1.8–2.1, or 260/230 ratio 2.0, was used to extract the RNA. The GoScript^®^ reverse transcription system (Promega, Madison, WI, USA) was used to reverse-transcribe a one microgram sample of RNA. The 20 L total volume of the RT-qPCR reaction samples was made up of 10 L of SYBR green master mix, 1 L of cDNA template, 8 L of ddH2O, and 1 L of each primer. A BIO-RAD CFX-96 device (Bio-Rad Laboratories, Inc., Hercules, CA, USA) was used to conduct RT-qPCR. The threshold cycle (Ct) value for each gene was normalized against the Ct value of ubiquitin and calculated relative to the relevant control samples as a calibrator using the equation 2-Ct. Each expression value was calculated as the average of three biological replicates and three technical replicates for each sample [177,178,179]. To distinguish between means for substantial effects, the standard error was used.

### 4.10. Analysis of Genetic Diversity in the K/Z RIL Population

The raw FASTQ reads from 200 genotypes comprising 2 parental genotypes and 198 lines of the K/Z RIL population were mapped to the reference rice genome cv. Nipponbare (IRGSP 1.0) using the Burrows–Wheeler Aligner (BWA) [180]. The bam files were used to call SNPs using GATK [181]. Approximately 6.5 million high-quality SNPs were retained (less than 2% missing rate and more than 5% minor allele frequency) and annotated. Of these, the SNPs present in the up- and down-stream region of the 26 drought loci were selected and analyzed further. Principal component analysis (PCA) was conducted using ‘--pca’ flag in PLINK v.1.9 [182] and visualized in R [183].

## 5. Conclusions

All morphological variables, components of grain yield, and root architecture traits revealed normal frequency distributions in the RIL population, indicating quantitative inheritance. A total of 11 lines from K/Z RIL population were identified as highly drought resistance based on the morphological traits and grain yield components under WW and DS conditions with root architectural traits under ABA conditions, including 100005, 100032, 100034, 100133, 100135, 100162, 100169, 100242, 100295, 100334, and 100337. Meanwhile, 10 highly drought sensitive were found, such as 100092, 100164, 100170, 100222, 100237, 100239, 100259, 100263, 100324, and 100345. In addition, ABA-mediated cell signaling aids drought-tolerant rice plants maintain grain yields under DS conditions by fostering cell elongation, preserving cellular membrane integrity, and regulating osmotic stress tolerance. All of these traits show a positive correlation with FG-DS, and this is also true for the major root architectural traits. Using QTL IciMapping, we analyzed QTL with a total of 4133 SNPs as markers. For traits affected by drought, a total of 41 QTLs and 184 candidate genes were found inside the QTL areas. The results of the RT-qPCR analysis showed that a large number of genes were up-regulated in the drought-resistant parent, Kaybonnet. Of these genes, 15 were candidates for drought resistance within QTL regions with known annotations, showing higher intrinsic values in Kaybonnet. Important for the development of drought-resistant rice varieties is a thorough understanding of the control of gene expression in response to drought stress, which can be gained by identifying candidate genes within the QTL areas.

## Figures and Tables

**Figure 1 ijms-24-15167-f001:**
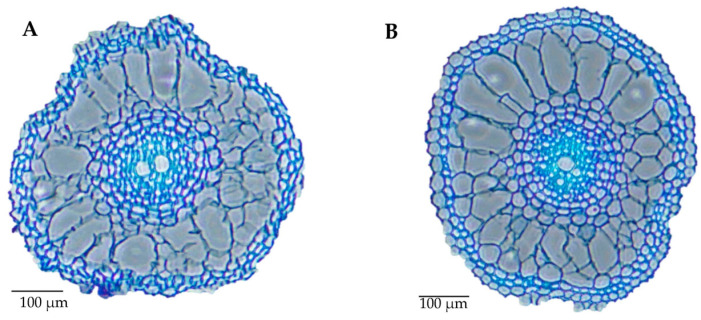
Root anatomy of parental lines Kaybonnet (**A**) and ZHE733 (**B**) at 40× magnification.

**Figure 2 ijms-24-15167-f002:**
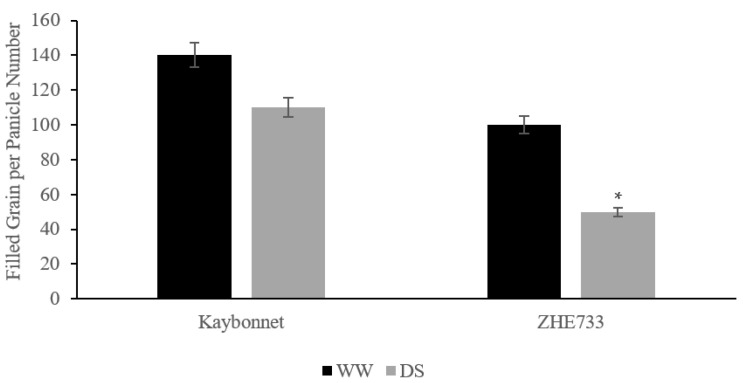
Number of filled grains per panicle in parental lines Kaybonnet and ZHE733. Kaybonnet maintained a higher number of filled grains under DS than ZHE733. * = significant using *t*-test *p* ≤ 0.01.

**Figure 3 ijms-24-15167-f003:**
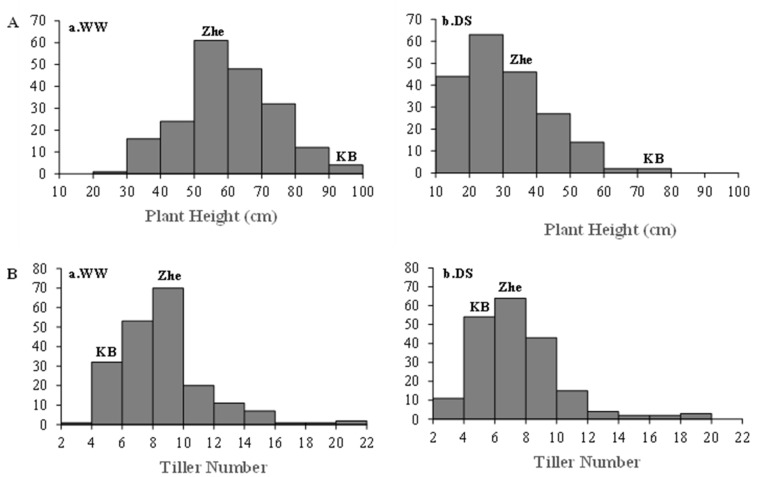
Frequency distribution of plant height (**A**) and productive tiller number (**B**) under WW (a) and DS (b) conditions.

**Figure 4 ijms-24-15167-f004:**
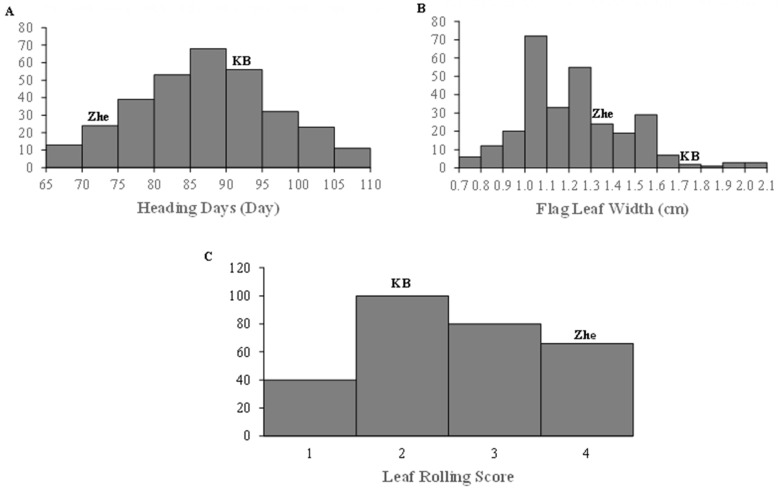
Frequency distribution of heading days (**A**), flag leaf width (**B**), and leaf rolling score (**C**).

**Figure 5 ijms-24-15167-f005:**
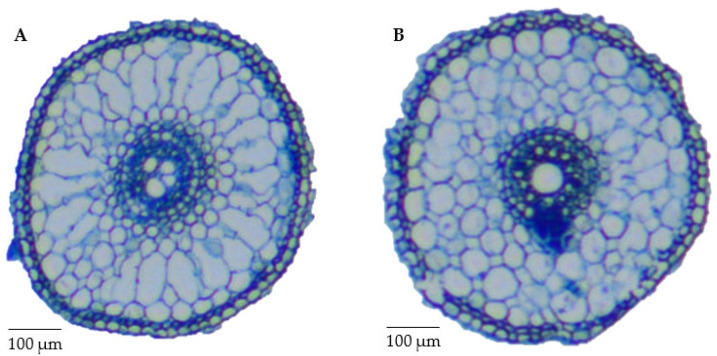
Root anatomy of 100162 as a drought-resistance line (**A**) and 100170 as a drought-sensitive line (**B**).

**Figure 6 ijms-24-15167-f006:**
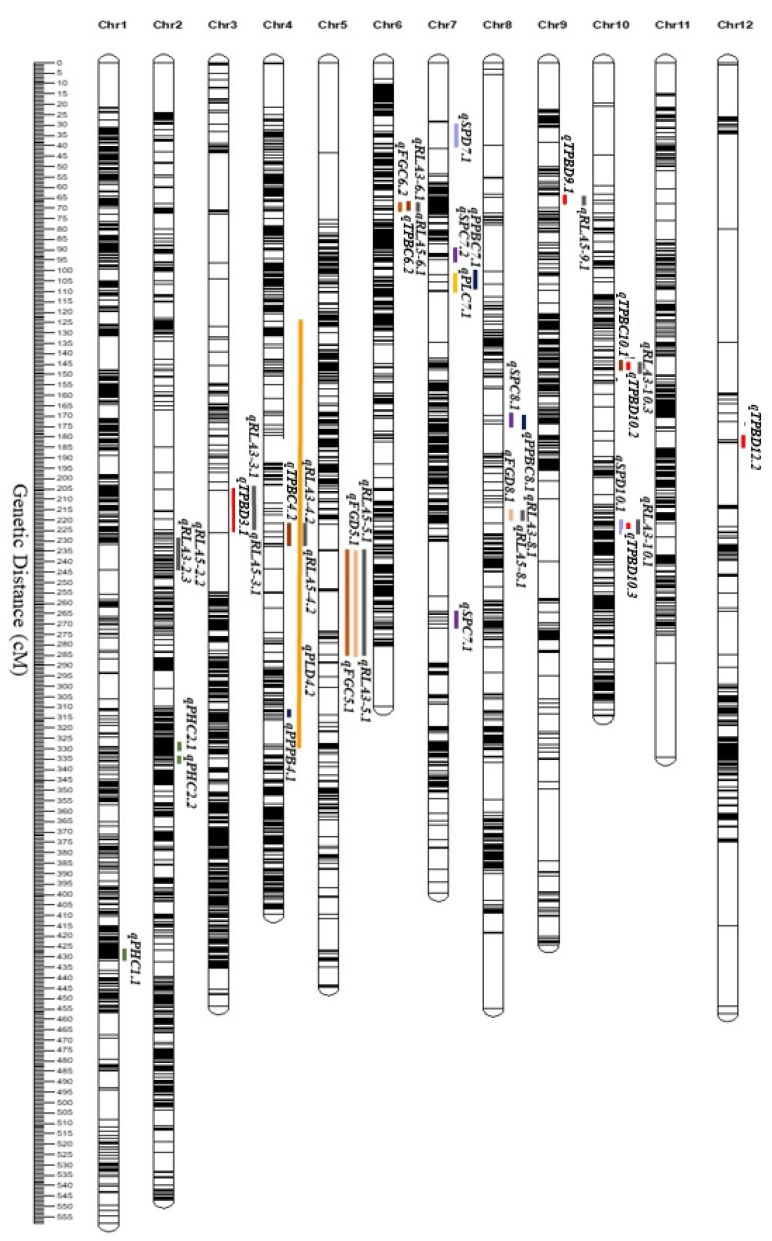
QTL locations of morphological and yield traits under DS and WW conditions for consecutive and combined years (2016, 2017, and 2018), and root architectural traits under ABA conditions in 2019 on the 12 rice chromosomes. The genetic distances (cM) are shown on the left of the chromosome (chr). The different colors represent morphological, yield, and root architectural traits under DS, WW, and ABA conditions.

**Figure 7 ijms-24-15167-f007:**
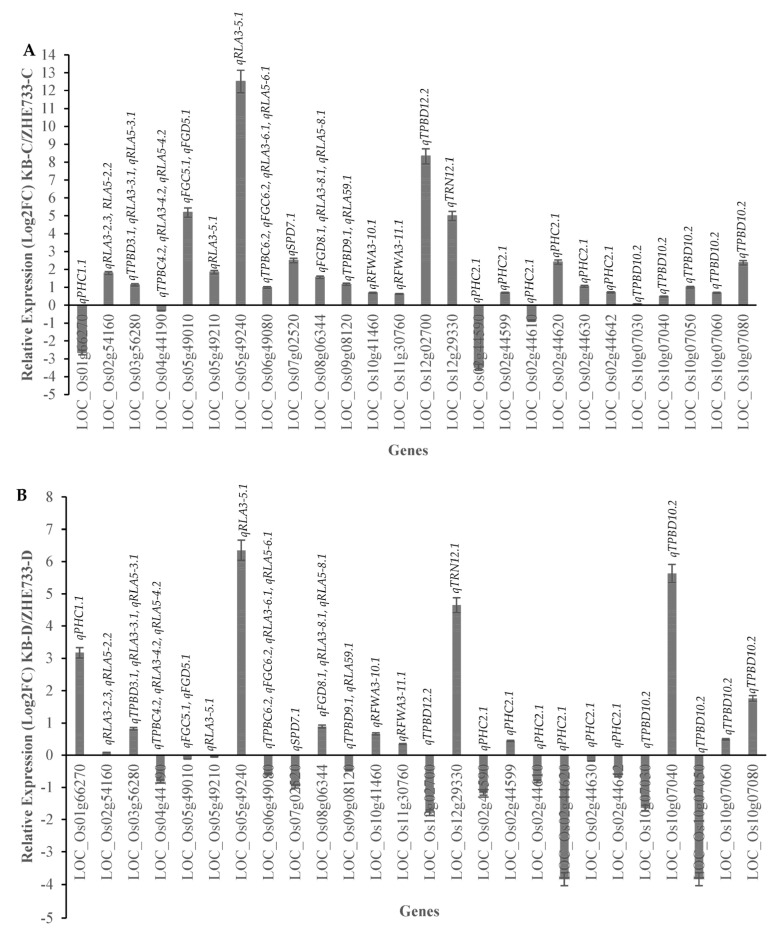
Expression profile of 26 candidate drought-resistance genes within QTL regions in leaf tissues of the parental K/Z RIL population, (**A**) Kaybonnet under WW conditions (KB−C) relative to ZHE733 under WW conditions (ZHE733−C), (**B**) Kaybonnet under DS conditions (KB−D) relative to ZHE733 under DS conditions (ZHE733−D).

**Table 1 ijms-24-15167-t001:** The average and range values of grain yield components of K/Z RIL population under WW and DS conditions.

Traits	Treatments	Parent				K/Z RIL Population		
		Kaybonnet	Reduction (%) Kaybonnet under DS	ZHE733	Reduction (%) ZHE733 under DS	Average	Reduction (%)	Range
Total plant biomass	WW	14.80 ± 0.11	4.73	13.40 ± 0.19	51.49	20.00 ± 0.24	12.50	4.00–49.00
	DS	14.10 ± 0.17		6.50 ± 0.24		17.50 ± 0.24		2.00–47.00
Spikelet per panicle number	WW	104.00 ± 0.22	28.85	90.00 ± 0.27	38.89	133.35 ± 0.31	38.75	45.00–328.00
	DS	74.00 ± 0.15		55.00 ± 0.16		81.68 ± 0.27		26.80–188.40
Filled grain per panicle number	WW	66.00 ± 0.23	24.24	43.20 ± 0.18	62.96	43.07 ± 0.21	47.62	2.00–176.00
	DS	50.00 ± 0.28		16.00 ± 0.12		22.56 ± 0.24		0.00–97.00
Panicle length (cm)	WW	21.70 ± 0.16	11.06	18.60 ± 0.15	16.13	21.24 ± 0.16	15.82	11.90–33.30
	DS	19.30 ± 0.12		15.60 ± 0.23		17.88 ± 0.19		11.64–26.62
Primary panicle branch number	WW	19.00 ± 0.11	31.58	8.00 ± 0.14	12.50	11.17 ± 0.13	16.92	3.00–22.00
	DS	13.00 ± 0.17		4.00 ± 0.12		9.28 ± 0.12		5.40–15.60

## Data Availability

Not applicable.

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
