# Peer review of "QTLs and Candidate Loci Associated with Drought Tolerance Traits of Kaybonnet x ZHE733 Recombinant Inbred Lines Rice Population"

_ijms, 2023, doi:10.3390/ijms242015167_

Round 1

Reviewer 1 Report

This manuscript describe a routine QTL mapping and candidate gene identification work, targeting rice drought-related traits.

There are two major drawbacks for this work: 

First, the authors claimed that the phenotyping work has been performed in 2016, 2017, 2018. However, they did not show the QTL identification (QTL map) for each year and comparison between the stable QTL regions identified among years. Then the downstream analysis for identifying candidate genes were challenging. 

Second, according to the current requirements for a rice QTL study, this work fall short in the following aspects, especially when considered to be published in a multi-disciplinary journal with a ever-increasing criteria (liek IJMS): (1) whole-genome sequencing is preferred for only 198 individuals, as the sequencing costs of NGS Illumina has becoming lower and lower; (2) functional validation using transgenic approaches for some of the candidate genes are expected, but none of the identified candidate genes came with functional studies! (3) Explanation or discussion on the different alleles between Kaybonnet and ZHE733 are expected to elaborate why some alleles could confer better drought-related phenotypes. 

Overall, this work presents a preliminary and routine results  of QTL mapping in rice. The depth of the work is not enough for consideration to get published in IJMS. The work also lacks detailed data mining and interpretation, as well as broad significance. So I suggest reject. 

Reviewer 2 Report

Drought is a most factor threatening rice yield anfd quality. In the present study, QTL analysis, candidate gene mining were donducted using a RIL population from Kaybonnet and ZHE733. A total of 41 QTLs and 26 candidate genes were identified. However, RT-qPCR of candidate genes were not enough for validation. The QTLs should be validated by other strategy, such as subsegregating populations. Further, the MS was not wrote in an logical methods. For example, the sections 2.1~2.6 should be combined and discribed more clearly. Thus, this MS is hard to understand.

I have no comments.

Reviewer 3 Report

Drought-resistant is important for rice production. This study has focused on QTLs and candidate loci associated with drought tolerance traits by using Kaybonnet x ZHE733 recombinant inbred lines. A total of 41 QTLs were identified. Before accepted, the manuscript should be addressed the following points:

1. In Background, it’s better to introduce more detail information of the QTLs and genes associated with drought resistant traits.

2. The data in supplementary table 1 as well as other tables should be showed in the form of “Mean ± SE”.

3. The author should annotate the results of the T-test on Figure 2.

4. Suggest the authors to merge Figure 3 and Figure 4 into one figure.

5. The authors should introduce more details about measurement of morphological traits and grain yield indicators in Materials and Methods. For example, how to divide the leaf rolling score (LR) into four levels? What are the criteria for classification?

6.In figure6, the authors should indicate what different colored lines represent.

7. In this study, 41 QTLs for root architectural traits under ABA treatments. Is this only one year results? Or the QTLs detected in all 3 years? It is not clear.

Minor editing of English language required

Reviewer 4 Report

Authors submitted manuscript entitled “QTLs and Candidate Loci Associated with Drought Tolerance 2 Traits of Kaybonnet x ZHE733 Recombinant Inbred Lines Rice 3 Population” to IJMS. The authors conducted a study on rice drought resistance using a cross between two rice varieties. They identified 41 key genetic regions and 184 candidate genes related to drought resistance and confirmed 17 of them, providing valuable insights for developing drought-resistant rice cultivars. The manuscript is written very well and reports good results. I feel that the whole manuscript is well balanced but the discussion needs some improvements.

Figures 1 and 2; improve figure legends. Add some explanation here.

The discussion is a little weak. The authors haven’t discussed the results in a critical manner. Moreover, some lines about gaps and perspectives are missing.

English is fine

Reviewer 5 Report

In this paper authors investigated genes responsible for drought tolerance in rice.  The paper represents solid analysis, well structured, the experiments are well designed and reliable. Conclusions are supported by the presented data.

The only minor critics relates mainly to typos and English usage:

A  normal  frequency  distribution – use just distribution, normal distribution is another thing.

while under drought stress the KB-DS (14.10g) showed significant decrease compared to ZHE733-DS (6.50g),  - must be vice versa

Section 2.5 .- not clear, must be better explained or removed. Please try to visually show the correlation.

If you mentioning correlation please elaborate on some traits as example  311

which strikes out.  - ???

typically  have  three  metaxylem  vessels – on the Fig 5a it is 4!

and  new  drought-resistant  rice  cultivars  were  developed. – are really developed here or can be developed?

Round 2

Reviewer 2 Report

The revised MS has been largely improved, and I have no commont anymore.

Reviewer 3 Report

 I feel the manuscript is well revised and it may be published. Finally, I endorse the publication.
